# Angular Scattering of the Sahara Dust Aerosol

Helmuth Horvath[1], Lucas Alados Arboledas[2,3], Francisco Jose Olmo Reyes[2,3]

[1]University of Vienna, Faculty of Physics, Aerosol Physics and Environmental Physics, A-1090 Vienna, Austria
[2]University of Granada, Department of Applied Physics, E-18071 Granada, Spain
3Andalusian Institute for Earth System Research (IISTA-CEAMA), Granada, Spain

*Correspondence to*: Helmuth Horvath (Helmuth.Horvath@univie.ac.at)

**Abstract.** Soil erosion aerosols can be transported considerable distances, the Sahara is one of the major sources on the world. In June 2016 the volume scattering function of the atmospheric aerosol has been determined in the Sierra Nevada, Spain, at an altitude of 2500 m. Measurements were performed with a polar nephelometer permitting measurements
between scattering angles of 5° to 175°. The values at the missing angles could be estimated to a high accuracy, using the shape of the scattering function adjacent to the missing angles, thus a complete volume scattering function was available. During the measuring period intrusions of long range transported Sahara aerosol happened several times. The classification of the aerosol was done by back trajectories and by the Angström exponent of the wavelength dependent scattering coefficient, which was determined by a three wavelength Integrating Nephelometer. The phase function of the Sahara
aerosol had a stronger forward scattering, and less backscattering compared to the non-Sahara aerosol, which is in agreement with other findings for irregular particles. The asymmetry parameter of the phase function is the best characteristic to distinguish Sahara Aerosol from non-Sahara aerosol. In this study the asymmetry parameter for the Sahara aerosol was larger than 0.65, whereas the non-Sahara aerosol had an asymmetry parameter below 0.6. A comparison with measurements performed with long range transported Gobi Desert aerosols observed in Kyoto, Japan, showed very similar results.

## 1. Introduction

Deserts are a major source of aerosol particles. On a global scale the desert aerosol contributes 60–1800Tg/y of the total yearly aerosol production of 2900–4000 Tg (Jaenicke, 1988). Junge (1979) estimates the global source strength of deserts as
260 to 400 Tg/y, with the Sahara contributing 60 to 200 Tg/y. IPCC (2001) estimates the yearly mineral dust emissions as 1000 to 3000 Tg/y, amounting to 1/3 to ½ of the global emissions. Recently the toal PM10 (particles smaller than 10 µm) emission of desert dust particle emissons is estimated as 1700 Tg/y, the atmospheric loading of dust particles PM10 is estimated as 20 Tg (Kok et al, 2017). For PM 20 (particles < 20µm) the values are 3000 Tg/y and 23 Tg. The clay sized particles (diameter smaller 2 µm) are an important subgroup, due to their efficient light scattering but account only for 3.5 to
5,7% of the PM10 emissions,

In contrary to most of the other particles present in the atmosphere, desert aerosol particles are produced from minerals by mechanical processes, which obviously lead to irregular shaped particles. This is documented by innumerous electron micrograph studies (see e.g. Falkovich et al. 2001, Iwasaka et al., 2003, Kandler et al, 2006, Kandler et al. 2011). The mechanical production of particles results in particle sizes usually larger than 1 µm. Mean sizes range up to 10µm (Kontratyev et al. (2006), Alfaro et al. (2003), Cheng et al. (2005), Xin et al. (2005), Alfaro & Gomez, (2001), Falkovich et al., (2001)), and during atmospheric transport the size distribution is modified, mainly due to the shorter life time of larger particles: The average residence time in the troposphere is estimated as 10 days for 1 µm particles and 3 days for 10 µm particles (Jaenicke, 1988). Another estimate of the life time of the size dependent desert particles is 11 days for 1µm and 0.4 days for 15µm (Kok et al 2017). Obviously the particle size distribution is modified, when transported in the atmosphere, since larger particles have a considerable sedimentation velocity, Therefore the clay sized particles amount to 15% of the global atmospheric load, although accounting only for 4.3% of the emitted mass. With the global wind system the smaller Sahara aerosol particles can be transported considerable distances, have been observed in the Amazonas basin (Formenti et al. 2001) and are a major source of minerals' supply to the Amazonas or Congo Basin (Okin et al., 2004).

Among many other effects, the desert aerosol is expected to have an influence on the radiative balance, see e.g. Obregón (2014), Valencuela et al. (2012 a,b), Antòn et al. (2014). Due to the irregular shape of the particles, they have a larger surface to volume ratio, leading to a higher extinction efficiency compared to volume equivalent spherical particles (Kocifaj and Horvath (2005), Kok et al (2017)). The maximum efficiency is around an equivalent diameter of 1 µm, thus, the clay sized particles are especially important. Recently it has been found, that the cooling effect of desert aerosol might be smaller than expected (Kok et al, 2017): At the source region coarse dust particles (>5µm) dominate and they absorb both solar and thermal radiation, furthermore the effect is augmented by the bright surface of the desert; distant from source regions the cooling effect of the fine dust seems to be less as previously anticipated.

The dust particles have an irregular shape, thus their scattering properties are difficult to model. If the shape, size, and possible inhomogeneities of the particles are known, the scattering matrix of the particle could be calculated (Lindquist et al., 2014). Since it is not possible to have an electron micrograph of every particle, simplifications are used: Lindquist et al. (2014) have considered the following models: homogeneous particles of irregular shape, Gaussian random spheres, spheroids and spheres. Whereas the scattering properties of homogeneous irregular particles do not differ much from reality of inhomgeneous particles, the other models give considerable deviations, e.g. when assuming spheres instead of irregular chrysotile particles the asymmetry parameter is 0.66 instead of 0.82, and the lidar ratio is 4 instead of 48 sr[-1].Obviously this discrepancy cannot be tolerated, Thus several indirect methods are in use, such as inverting sun and sky radiation measurements (e.g. Estelles et al. 2007), or using measured size distributions and Mie Theory to calculate the scattering, or

using an intergrating nephelomter with two scattering ranges and Henyey Greenstein model phase function (Andrews et al 2006). Always model assumptions have to be made in order to obtain the complete scattering properties of the aerosol containing irregular shaped particles, which may create considerable uncertainties as shown above. But the exact scattering properties, especially the asymmetry parameter, are urgently needed to estimate the effect on climate. Below we report the

first direct measurements of the complete volume scattering function of the Sahara aerosol transported to the Iberian Peninsula, which were performed during several dust outbreaks. No assumption or model is needed in order to obtain the volume scattering function and the asymmetry parameter.

## 2 Definitions, units, and nomenclature

The volume scattering function $\gamma(\theta)$ of the aerosol is defined as follows, see figure 1: Let a volume $dV$ of aerosol be

illuminated by radiation having a flux density S. The light flux $d\Phi$ scattered into a solid angle $d\omega$ at the scattering angle $\theta$ is obtained as

$d\Phi = S. \gamma(\theta) . d\omega . dV.$    The unit is $[\gamma] = m^{-1}.sr^{-1}$.

The total scattering coefficient $\sigma_s$ is obtained by integrating the volume scattering function $\gamma(\theta)$ over whole solid angle:

$\sigma_s = {}_0\int^{4\pi} \gamma(\theta) \, d\omega,$

or for scattering with rotational  symmetry with respect to the incident beam

$\sigma_s = 2\pi._0\int^{\pi} \gamma(\theta) \sin(\theta) \, d\theta.$

The scattering coefficient can be understood as the fraction of the light flux scattered per unit length out of a parallel beam of light, its unit is $[\sigma_s] = m^{-1}$. Both the volume scattering function and the scattering coefficient are extensive properties.

The phase function is an intensive property and describes the relative angular dependence of the scattered light of a volume element of particles. For the phase function $P(\theta)$ we have used the following definition:  $P(\theta) = 4.\pi.\gamma(\theta)/\sigma_s$ .

The angular distribution of the scattered light frequently is characterized by two parameters:

The asymmetry parameter g is obtained by folding the phase function with cos $(\theta)$, therefore

$g = \frac{1}{2} {}_0\int^{\pi} P(\theta) \sin(\theta) \cos(\theta) \, d\theta.$

The asymmetry parameter is zero for symmetric scattering such as Rayleigh scatter of the air molecules and g=1 for only forward scatter. Another characteristic is the fraction b of the back scattered radiation. It is obtained by integrating ½ $P(\theta) \sin(\theta)$  between ½$\pi$ and $\pi$. Both parameters g and b are intensive.

The lidar ratio, S, is defined as the ratio of the extinction coefficient, $\sigma_e$  and the volume scattering function at 180°, $\gamma(180°)$, i.e. S= $\sigma_e/ \gamma(180°)$, or S= $4\pi /([P(180°) .\omega]$, with $\omega$ the single scattering albedo, i.e. the ratio of the scattering coefficient to

the extinction coefficient. The single scattering albedo cannot be measured directly with the polar nephelometer. We have used values for $\omega$, which were obtained by inverting data from sun and sky photometers during this study. The average for

the Sahara aerosol was $\omega = 0.928$ and $\omega = 0.943$ for the non-Sahara aerosol. These values are in agreement with previous findings (Valenzuela et. al. 2012a,b)

The measurement of the scattering coefficient at three wavelengths (e.g. with an integrating nephelometer, Charlson et al., 1967) permitted the determination of the wavelength dependence of the scattering coefficient. In many cases it can be represented by a power law relation $\sigma(\lambda) = \sigma(\lambda_0) \cdot (\lambda/\lambda_0)^{\alpha}$ with $\alpha$ the Ångström exponent (Ångström 1929, 1930). The value of $\alpha$ is independent of the absolute magnitude of the scattering coefficient, i.e. the concentration of the particles. But it is different for different types of aerosols and mainly influenced by the size distribution of the particles. For a power law number size distribution given by $dn/dr = n_0 \cdot (r/r_0)^{-\nu}$, the exponent $\alpha$ can be obtained by $\alpha = 3 - \nu$. (Junge 1963). For an aerosol with a size distribution having $\nu > 3$, the smaller particles dominate and $\alpha < 0$, i.e. the scattering coefficient decreases with increasing wavelength, which is the normal case. If the particles are larger than a few micrometers, then $\nu < 3$ and the Ångström exponent is zero or even positive. A strong wavelength dependence of the scattering coefficient ($\alpha < -1$) is an indication for the non-Sahara aerosol, whereas little or almost no dependence on wavelength is a sign for an aerosol containing larger particles, in this study mainly desert particles.

## 3. Instrument and methods

The volume scattering function of the aerosol has been determined by a custom made polar nephelometer, its design is similar to the one of Waldram (1945), see figure 2: Light from a 532 nm solid state laser of 10 mW power shines into a light trap and illuminates the particles within the beam. A photomultiplier with a collimation optic is mounted on a goniometer and can measure scattered light between 5 and 175°. The scattering volume is approximately 70 mm$^3$ at 90° scattering angle, increasing to 800 mm$^3$ at 5° and 175° respectively. Measurements are taken at intervals of 5°, in the near forward direction at 1 to 2°. For one volume scattering function a scan from 175° to 5° and back is made and the average is used. One full scan takes 35 minutes, which can pose a problem with rapidly changing aerosols. Therefore a check for differences between the forward and the back scan is made and if the difference was larger than a factor of 1.5 the data were not used. The instrument is enclosed in an airtight housing; by sucking air out of the enclosure, air from outside can be brought into the instrument. Calibration is done by filling the instrument with carbon dioxide, whose volume scattering function is well known. For all data reported below, the scattering of the air molecules has been subtracted, i.e. all scattering functions, phase functions, asymmetry parameters or back scattered fractions are for aerosol particles only.

Experimentally it is impossible to measure the volume scattering function between 0 to 5°, and 175° to 180°; but the contribution of this range cannot be neglected, especially the forward region. The measured volume scattering function and its shape between 10° and 5° and 170° and 175° can be used to extrapolate the missing regions. This can be done with an accuracy of better than 5% (Horvath, 2015), thus the complete scattering function is available.

The flow rate through the instrument was $3.3 \times 10^{-4}$ m$^3$ s$^{-1}$. The connection to the sampling inlet of the field laboratory was by a slightly downwards inclined hose of a diameter of 10 mm and a length of 2.5 m. Using data given by given by Hinds (1999, Chapter 10), the loss of particles due to sedimentation in the tube amounted to 10%, 2%, and 0.3% for particles with diameters of 10, 5, and 2 µm. Losses due to diffusion were below 0.02%. Thus it can be concluded, that for the most important particles sizes below 5 µm the sampling losses are negligible.

A TSI 3563 Integrating Nephelometer was used to measure the scattering coefficient for red (700 nm), green (550 nm), and blue (450 nm) light. A detailed description of the instrument and its operation can be found at the NOAA website https://www.esrl.noaa.gov/gmd/aero/instrumentation/neph_desc.html.

Sampling took place in the Albergue Universitario of the University of Granada, located in the Sierra Nevada at an elevation of 2505 m a.s.l. Its coordinates are 37° 5' 43.72"N, 3° 23' 12.57"W. The surrounding mountains had elevations of approx. 3000 m, extending at distances of 20 km from the sampling location. All instruments were located in a room below the flat roof of the building. A pipe with a diameter of 10 cm extended 2.1 m above the roof and into the laboratory. A blower maintained a flow of 0.00167 m$^3$s$^{-1}$, all instruments used in the SLOPE campaign (Integrating Nephelometer, Aerodynamic Particle Sizer (APS), Multi Angle Absorption Photometer (MAAP), Scanning Mobility Particle Spectrometer (SMPS), Aethalometer (A33)), and the polar nephelometer) sampled from this pipe. The residence time of the air in the pipe was 0.6 s. Losses in the pipe can be considered negligible.

Measurements were made in the framework of the SLOPE (**S**ierra Nevada **L**idar Aer**O**sol **P**rofiling **E**xperiment) campaign between June 6 and 30, 2016. This campaign mainly was intended to determine the vertical structure of the aerosol by remote sensing instruments and test the various retrieval schemes for obtaining microphysical and optical properties. So the main instruments were sun and sky photometers, multiwavelength lidar, and an airplane. Obviously ground based instruments were also used, as described above. The polar nephelometer could not be operated continuously, due to instrument failures and absence of the operator; still, in total 120 phase functions were determined. During this time several intrusions of Sahara dust occurred, usually the dust was layered and could be recognized with the unaided eye.[1] For distinguishing dust aerosols from others, two methods have been applied: (1) The size of the dust particles is larger than 1µm, thus the Angström exponent normally is larger than -1, whereas the non-Sahara aerosol has Angström exponents below -1. (2) Using 72 hour back trajectories (from the NOAA HYSPLIT web site, Draxler & Rolph, 2003), the likely origin of the

---

[1] From May to September 2016 there were a total of 15 Sahara dust events, on 96 days out of 153. For details see https://www.mapama.gob.es/es/calidad-y-evaluacion-ambiental/temas/atmosfera -y-calidad-del-aire/episodiosnaturales2016_tcm30-379284.pdf. In June 2016 the events were on the following days: 2-3, 6-11, and 21-30.

particles can be estimated. Since the Sierra Nevada is a small mountain range, most of the time the air masses reached the measuring location (at an elevation of 2500 m a.s.l.) without admixing of particles of possible nearby sources. The origin of the air mass was classified in a total of six groups. Figure 3 shows the typical situations: Sahara, Sahara high, Atlantic, Atlantic North, Atlantic/Sahara, North Africa/Mediterranean. From the source region in the Sahara to the receptor region in the Sierra Nevada the particles traveled around 1500 km. Table 1 characterizes the 6 types, which occurred during this campaign.

A plot of the Angström exponent as a function of time measured with the three wavelength integrating nephelometer is shown in Figure 4. The classification using the back trajectories is given above. It is evident, that for aerosols influenced by the Sahara, the Angström exponent is larger than -1, thus the two methods of determining Sahara aerosols agree in most of the cases.

## 4. Results

An overview of the scattering coefficient obtained by integration of the measured volume scattering function and the value measured with the Integrating Nephelometer is shown in figure 5, only the light scattering of the particles is plotted, i.e. the Rayleigh scattering of the air was subtracted. The data obtained with the Integrating Nephelometer are values, taken at intervals of minutes, and depict also the variability of the layered aerosol. The scattering coefficient obtained by integration of the polar nephelometer data are averages of about 35 minutes. Agreement between the two datasets is evident. Three periods of measurements can be seen. During Period 1 mostly a distinct intrusion of Sahara Aerosol was observed, except for the beginning, in Period 2 mainly aerosol from the Atlantic reached the site, whereas during Period 3 again Sahara particles dominated the aerosol. The air masses passing over the Atlantic had a much lower scattering coefficient, being about twice the one of pure air.

A plot of all Sahara and non-Sahara Phase functions is depicted in figure 6. There is some scatter in the data due to the layered aerosol, but it is evident, that the phase functions of the Sahara and non-Sahara aerosol are different. The averages of all phase functions of definite Sahara origin and of definite non-Sahara origin and the standard deviations are shown in figure 7.

## 5. Discussion

The comparison between the two types of average phase functions of figure 7 shows definite differences. The non-Sahara (mainly Atlantic) phase function has less forward scatter (on the average 31 sr$^{-1}$ at 0°) than the Sahara phase function (62 sr$^{-}$

[1]). This is readily explained by the larger size of the desert particles. Whereas spherical particles are a good approximation for the near forward scattering, the smaller backscattering of the Sahara phase function compared to the non-Sahara particles only can be explained by the irregular shape of the particles. For spherical large particles interferences and resonances are most pronounced, which leads to a considerable increase in backscattering and a low side scattering, which both have not been observed. For irregular shaped particles which furthermore are randomly oriented by Brownian rotation, the backscattering is by far less up to a factor of 10 compared to spherical particles, see e.g. Von Hoyningen-Huene and Posse (1997) or Mishchenko et al. (1997), Mishchenko (2000), Nousiainen and Kandler (2015).

The complete volume scattering function or phase function can be used to determine derived properties. The scattering coefficient is obtained by integrating the volume scattering function over the full solid angle. A comparison with the calculated and the measured scattering coefficient was shown in figure 5, the agreement is evident.

For modelling of e.g. radiative transfer, the asymmetry parameter of an aerosol is an important parameter. It is obtained by folding the phase function with the cosine of the scattering angle. For the average phase functions shown in figure 7 the asymmetry parameter for the Sahara aerosol particles is 0.71 with a standard deviation of 0.03, for the non-Sahara particles it is 0.56±0.04. The difference is significant, therefore the asymmetry parameter is a good indicator for the desert aerosol. In figure 4 the measured asymmetry parameter is added to the graph of the Angström exponent and the origin of the aerosol. Whenever the origin of the aerosol indicates desert particles the asymmetry parameter is high. At the same time the Angström exponent also is high. Figure 8 is a plot of all the data points and shows the relationship between Angström exponent and the asymmetry parameter for all measured phase functions for which Integrating Nephelometer data and polar nephelometer data were available simultaneously. Evidently a larger asymmetry parameter is associated with a larger Angström exponent, but the relationship is not very pronounced.

The scattering function of the desert aerosol has a low back scattering, which is typical for non-spherical particles. Therefore it is to be expected, that a characterization of desert aerosol particles could be achieved by considering the fraction of backscattered light. It is defined as the ratio of integral of the volume scattering function between 90 and 180° divided by the integral over the full angle and is readily available once the volume scattering function is known. A time series of the backscattered fraction obtained in this way is shown in Figure 9, black squares; The back scattered fraction obtained from polar nephelometer measurements is lower for the aerosol dominated by desert particles, as expected; but less distinct than the asymmetry parameter. Furthermore the back scattered fraction obtained with the Integrating Nephelometer (red line) is systematically larger, which has the following reason: The backscattered fraction for the Integrating Nephelometer is obtained by dividing the signals BbsG (signal of the Nephelometer in the backscattering range for green light) by BsG (signal of the total scattered light). Both signals are proportional to the light flux scattered by the aerosol, but they are truncated, since it is experimentally impossible integrate the scattered light flux from 90° to 180° and 0° to 180° respectively.

But the measured volume scattering functions permits the simulation of the truncation effect; and for a range of the scattering angles between 8° and 170° the simulated BbsG is obtained by integrating the measured volume scattering function from 90 to 170° and BsG is obtained by integrating the measured volume scattering function from 8 to 170°. The simulated truncated backscatters fraction is calculated by dividing the simulated Bbsg by the simulated BsG. The expected signal is also shown (blue circles), which is in much better agreement with the Integrating Nephelometer data.

For the Sahara-particle dominated aerosol the asymmetry parameter is larger and the back scattered fraction is smaller than for the non-Sahara aerosol. So it is obvious to use both parameters to characterize the Sahara aerosol. A plot of all data points in the (b,g) plane is shown in figure 10. The points representing the Sahara aerosol and the non-Sahara aerosol are well separated. The curve gives the relationship between the backscattered fraction and the asymmetry parameter calculated for monomodal spherical particles. For other refractive indices or ellipsoidal particles an almost identical curve is obtained, the point of inflection is at a slightly different location. For bimodal size distributions the data points are located to the right of the curve (Horvath et al., 2016), as it is the case for the data of this study. A clear distinction between the points representing Sahara and non-Sahara particles is possible. Additionally, points are plotted, which were obtained in Kyoto, Japan, during an event of long range transport dust intrusion from the Gobi Desert. These data perfectly fit together with the Sahara data.

## 6. Conclusion

The volume scattering function of the atmospheric aerosol was measured in the Sierra Nevada, where intrusions of Sahara aerosol are frequent. The origin of the aerosol particles was determined by back trajectories and/or by the Angström exponent of the wavelength dependence of the scattering coefficient of the aerosol. Usually it took 48 hours or more from the origin to the measuring site. Thus particles larger than 8µm have been lost by sedimentation and the remaining coarse dust particles were reduced considerably.

The phase function of the Sahara aerosol has more forward scatter, and less back scatter; it is more asymmetric than the non-Sahara aerosol, which in this study was mainly marine, with little continental influence. A few parameters of the aerosol are listed in table 2.

The best distinction between Sahara and non-Sahara aerosol is possible when using the asymmetry parameter, as is evident also in figures 5 and 11. This study suggests that particles causing an asymmetry parameter of the phase function above a value of 0.65 could be considered of Sahara desert origin. The asymmetric scattering of the Gobi desert aerosol is very similar to the one of the Sahara Aerosol.

The asymmetry parameter can be used as a distinction between aerosol dominated by desert particles and the other aerosol particles. In this work the desert dominated particles have an average asymmetry parameter of g=0.71, whereas the mainly marine aerosol has g=0.56. The asymmetry parameter of the aerosol measured in Vienna is below 0.65. This aerosol is little influenced by maritime and not at all by particles of desert origin, it could be considered as continental background plus traffic related particles (Horvath et al., 2016). The PM10 is regulated in Europe's air quality standards to a 24 hr average of 50µg/m3 with 35 permitted exceedances per year, and a yearly average of 40 µg/m3. Frequent dust intrusions might make it impossible to comply with the regulations, e.g. in 2016 Sahara dust intrusions in the vicinity of the Sierra Nevada were on 96 days. But natural sources are allowed to be excluded from the rules governing the air quality standards, if their origin can definitely be proven.   In that case the measurement of the asymmetry parameter possibly in combination with the backscattered fraction definitely will create clarity.

**Acknowledgements**

This work was supported by the Andalusia Regional Government through project P12-RNM-2409, by the Spanish Agencia Estatal de Investigación, AEI, through projects CGL2016-81092-R and CGL2017-90884-REDT. We acknowledge the financial support by the European Union's Horizon 2020 research and innovation program through project ACTRIS-2 (grant agreement No 654109). The authors thankfully acknowledge the FEDER program for the instrumentation used in this work and the University of Granada that supported this study through the Excellence Unit Program.

*Author contributions.*  L.A. and F.J.O. planned the SCOPE project, its financing and supplied the infrastructure in the Sierra Nevada site, took care of the operation and data analysis of the integrating nephelometer as well as many other instruments, and supplied data on dust events. H.H. designed and operated the polar nephelometer, reduced the data and performed the data analysis, wrote the draft and after through discussion with all authors finalized the paper.

*Competing interests.*  The authors declare that they have no conflict of interest.

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

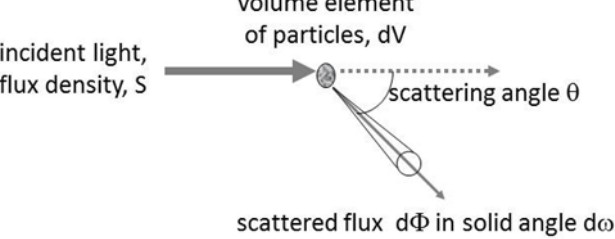

Figure 1: Definition of volume scattering function

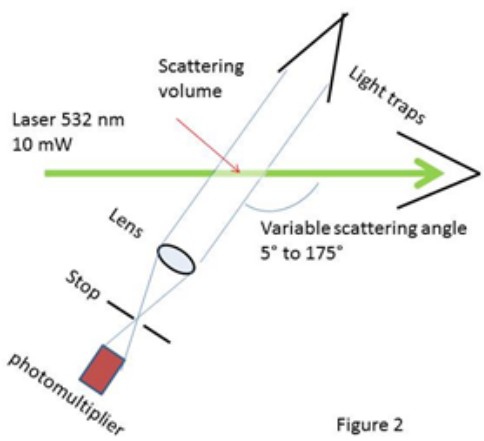

Figure 2

10   Figure 2: Principle of the polar nephelometer

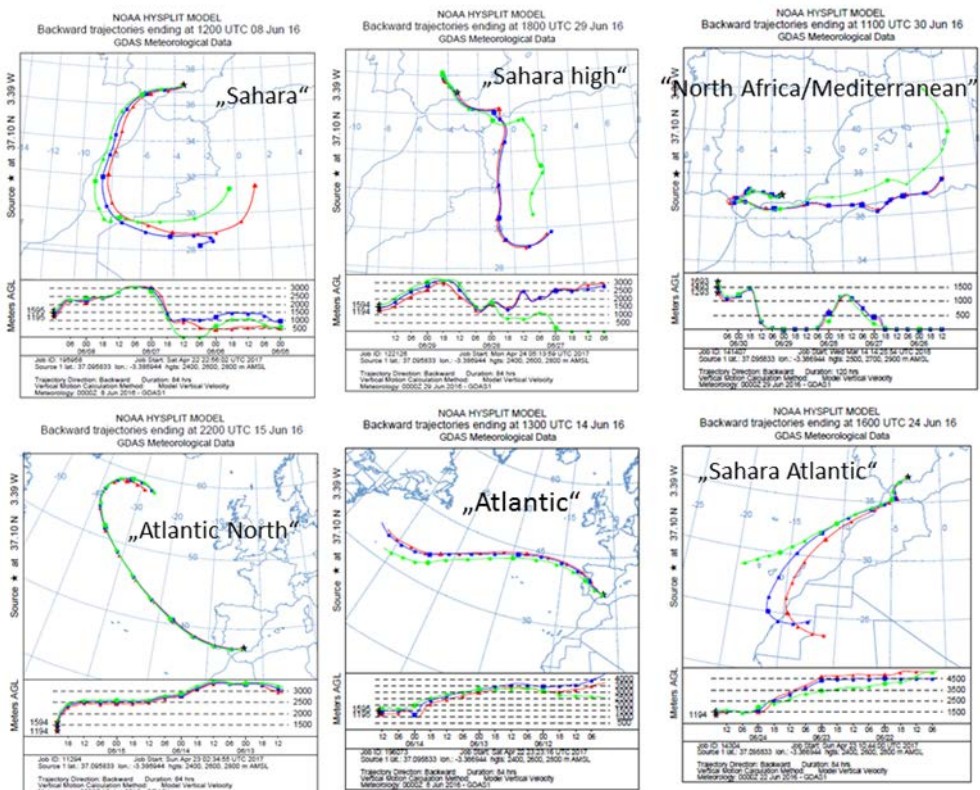

Figure 3

Figure 3: Back trajectories used for classification of aerosol types

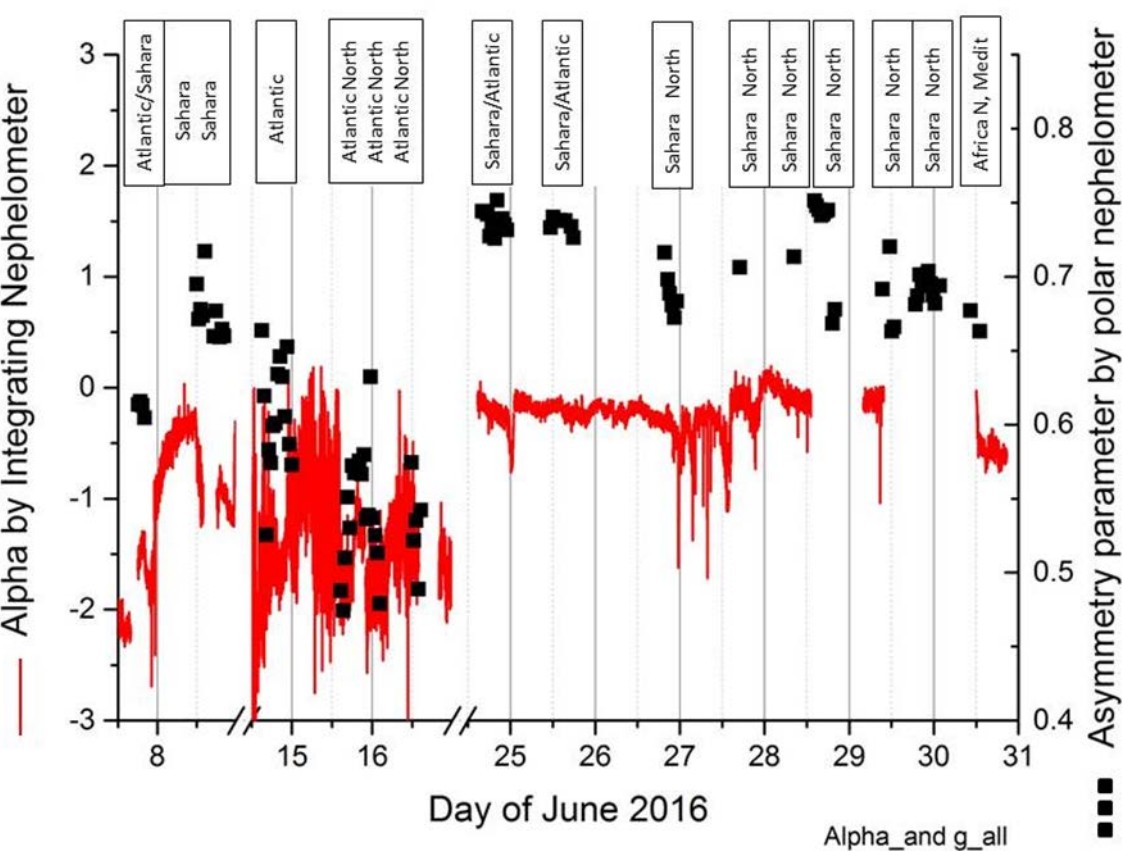

Figure 4: Angström exponent of the spectral scattering coefficient of the aerosol measured and classification by back trajectories. For the desert aerosol the Angström Exponent is larger than -1. In addition the asymmetry parameter is plotted too, for desert aerosol it is larger than 0.65.

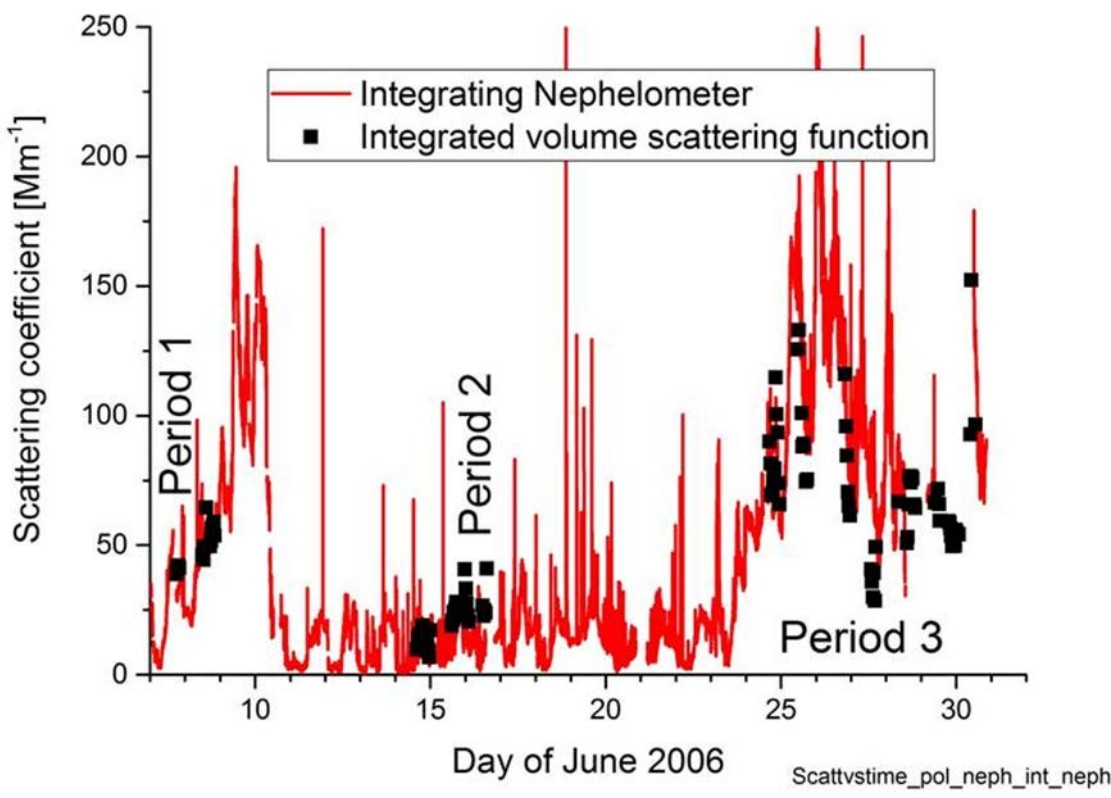

Figure 5: Measured scattering coefficient of the aerosol during the three periods of observation. The solid red line is the signal of the Integrating Nephelometer, the points are scattering coefficients obtained by integrating the measured volume

10    scattering function.

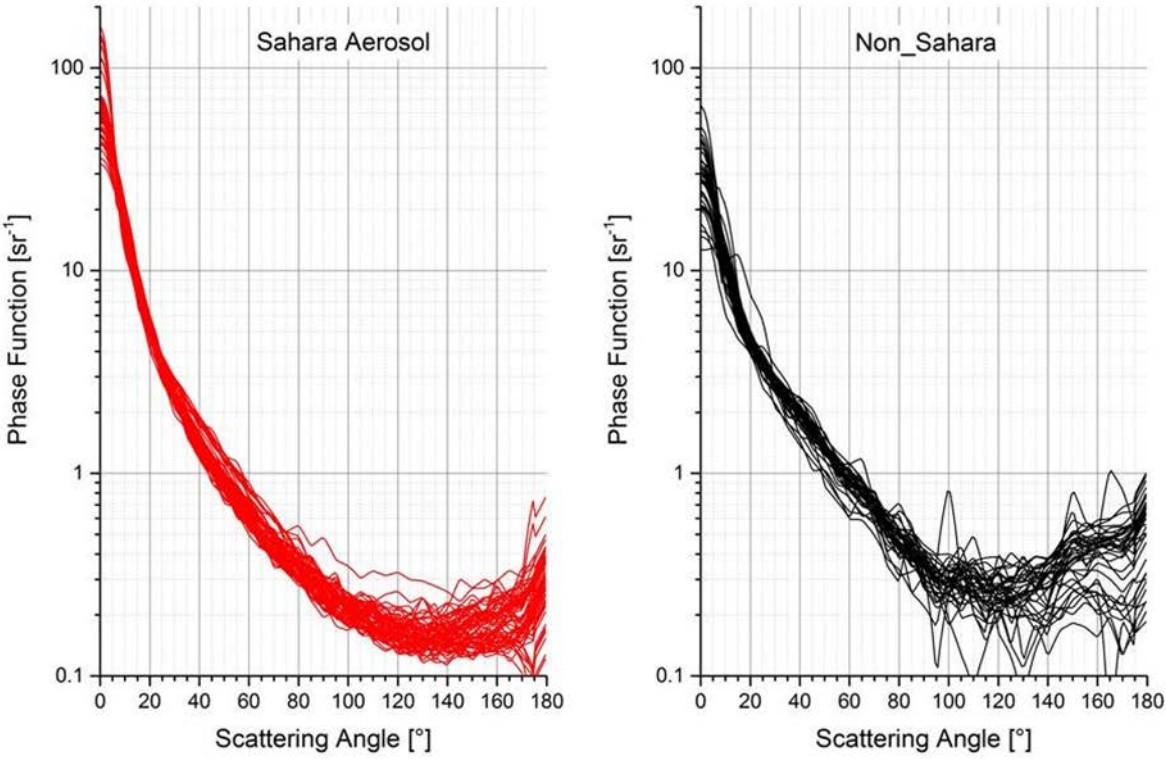

Figure 6. Comparison of phase function attributed to Sahara and non-Sahara aerosols.

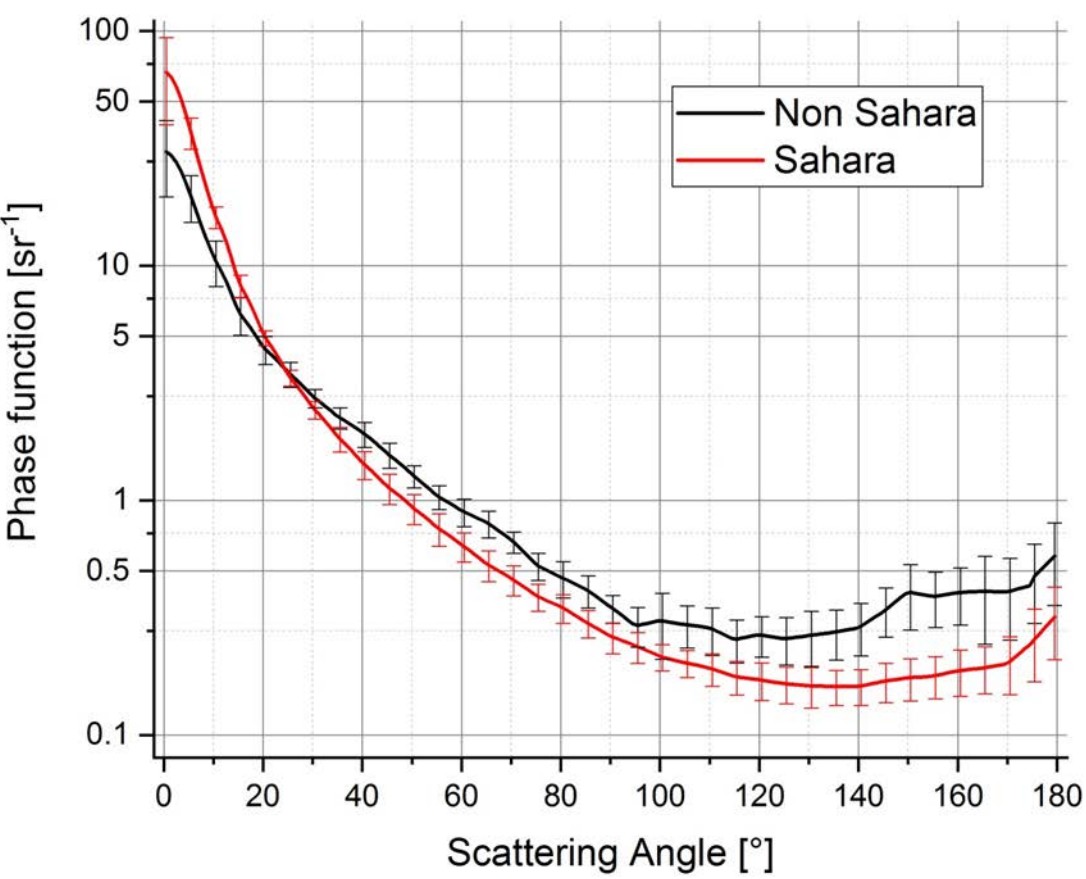

Figure 7. Average of the Sahara and non-Sahara aerosol phase functions.

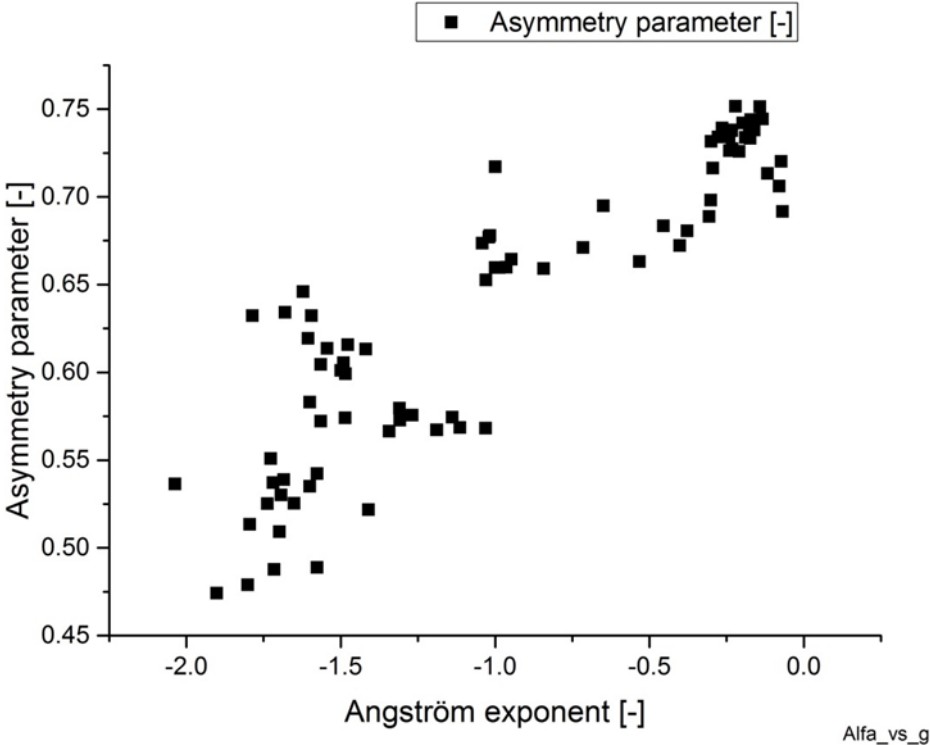

Figure 8. Relationship between the measured Angström exponent and the asymmetry parameter using the measured phase function.

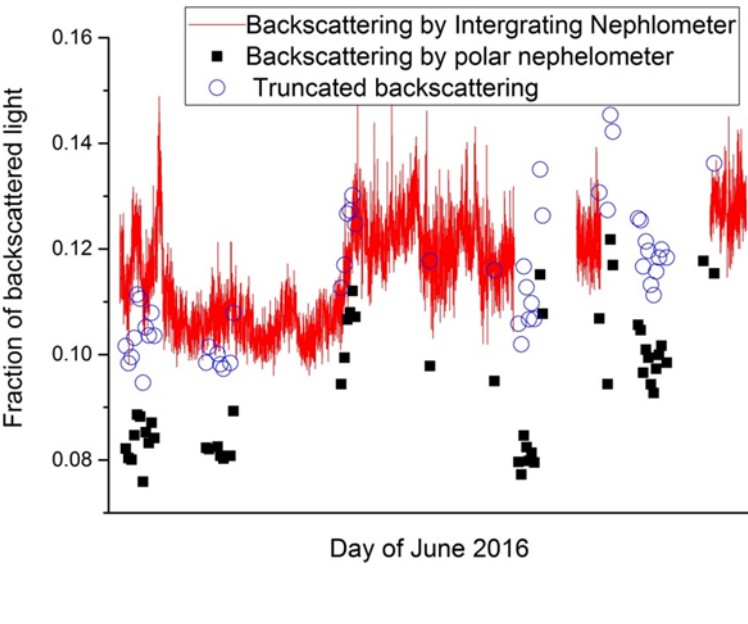

Back_period3

Figure 9. Back scattered fraction measured with the Integrating Nephelometer (red curve) and values obtained by the
measured Volume scatteing function. Simulation of the truncation is shown by the hollow points.

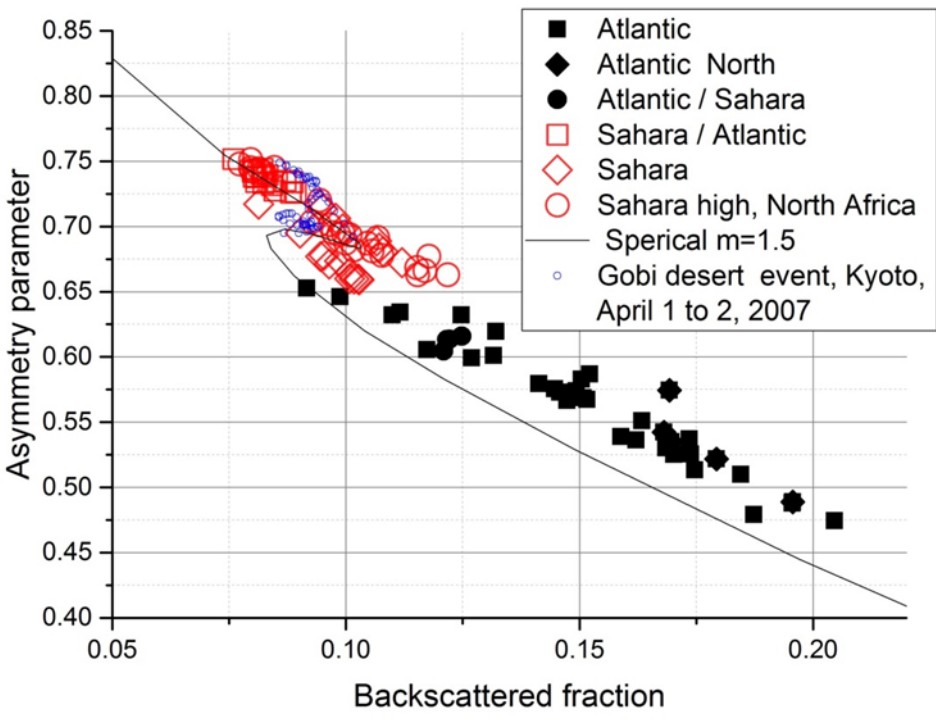

AsybackAerosolTyps_with_Gobi

Figure 10: Asymmetry parameter and backscattered fraction calculated for the aerosols measured during this study. A clear separation between Sahara and non-Sahara aerosols is evident. The data points are in excellent agreement with data for the

5   Gobi Desert aerosol.

| Table 1. Classification of back trajectories | | |
|---|---|---|
| Name | Air passes over | Remark |
| „Atlantic" | 60°W<longitude<0°, 35°N<latitude<40°N | |
| "Atlantic North" | 40°W<longitude<0°, 60°N<latitude<40°N | |
| "Sahara" | 10°W<longitude<5°E,37°N<latitude<26°N | |
| "Sahara high" | 5°W<longitude<5°E, 37°N<latitude<26°N | |
| "Sahara/Atlantic", "Atlantic/Sahara" resp. | Up to 500 km West of the coast of Africa | Distinction between the two by Angström exponent |
| "North Africa/Mediterranean" | 6°W<longitude<8°E, 35°N<latitude<37° | |

| Table 2. Characteristics of the two types of aerosols | | | | |
|---|---|---|---|---|
| | Sahara aerosol | | Non-Sahara Aerosol | |
| | Value | Stddev. | Value | Stddev. |
| Asymmetry parameter  [-] | 0.71 | 0.03 | 0.56 | 0.05 |
| Backscattered fraction  [-] | 0.094 | 0.014 | 0.153 | 0.027 |
| Average scattering coefficient [Mm$^{-1}$] | 71 | 22 | 22.3 | 7.3 |
| Phase function at 0°  [sr$^{-1}$] | 67 | 27 | 31 | 10 |
| Phase function at 90°  [sr$^{-1}$] | 0.26 | 0.03 | 0.35 | 0.04 |
| Phase function at 180°  [sr$^{-1}$] | 0.31 | 0.11 | 0.59 | 0.21 |
| Lidar ratio [sr$^{-1}$] | 44 | 16 | 23 | 8 |

