# Peer review of "Angular Scattering of the Sahara Dust Aerosol"

_Atmospheric Chemistry and Physics, 2018_

## Referee Comment (RC1) · Anonymous Referee #1 · 17 Jul 2018

First, I apologize for the delay of my review.

The paper provides measurements of the aerosol phase function from a custom made polar nephelometer for atmospheric aerosols observed at a high altitude site in Spain. The measurement period covers the month of June 2006 during which different cases of aerosols transport from the Atlantic Ocean, the Mediterranean, and the Sahara desert were measured. Dust and non-dust aerosol cases were distinguished based on the Scattering Angstrom Exponent and backtrajectory analysis. Results confirm previous observations that dust aerosols have larger forward and side scattering and less backscattering compared to non-dust cases. The asymmetry factor and its relation to the backscatter fraction is also investigated for the different aerosol observations. The paper provides valuable measurements of the phase function of aerosols of different origins and types, which is of interest for remote sensing and climate applications. Nonetheless, the paper has several weaknesses, both in terms of the presentation

form and discussion of the obtained results. Detailed comments are provided in the following. My recommendation is major revisions.

General comments

Introduction: The introductory part is quite poor and references are in part outdated. For example, numbers provided in the first three lines of the introduction are from a paper published 30 years ago. I suggest to revise the numbers on aerosol emission based on more recent literature. Also, more in general, I would suggest to better place your work in the literature context. In particular: I suggest to discuss briefly the state of the art and uncertainties on the phase function estimates for dust particles, difficulties related to its modelling and measurements, and the implications related to the improvement of its estimate.

Air-mass classification: I have a concern about the representativeness of the data for dust cases. Measurements are done on atmospheric aerosols classified as Saharan or non-Saharan based on the Angstrom exponent and trajectories. However, the possible mixing of dust with other aerosol types cannot be excluded in principle. This should be quantified or at least discussed. Also, what about the local aerosol contribution? I would suggest to compare the phase function obtained for dust aerosols in this paper with the one estimated by previous studies, possibly on pure dust, in order to assess possible differences and similarities and link them to atmospheric conditions.

In general, the paper misses from a formal uncertainty analysis of measured and derived parameters. Also, uncertainties are missing in all plots. Please correct this and add more formal error discussion. For example, in Figure 7 of the paper we can see differences between the dust and non-dust cases, but what is the real difference within uncertainties?

Specific comments

Page 1, line 29: please check the extra comma in the text

Page 2, line 4: add references after "larger particles"

Page 2, line 12: E should be replaced by S, I guess

Page 2, line 30: please check the formula since I think you missed a minus sign before the angstrom exponent; if I am right, then check the following discussion

Page 3, line 5: I do not like the expression "usual aerosol", please be more specific (pollution aerosols, fine aerosols?)

Page 3 line 5; I would replace with "is a sign for desert aerosol particles" with "it is a sign for large aerosols, as desert dust" or similar

Page 3, line 25: how the extrapolation is done?

Page 4, line 5: you refer to "all the instruments", which instruments? Please describe clearly the instruments used. I am also a bit confused by the fluxes. A flux of aspiration for the custom made nephelometer is specified in the previous page, while here there is reference to a different flow rate. What is this for?

Page 4, line 12: again there is the expression "usual aerosol" to modify

Page 4, line 19-22: the integrated nephelometer mentioned in this paragraph was not introduced before. Please, again, clearly indicate the used instruments and their configuration. Moreover, what about the integrating nephelometer (model, data treatment, uncertainties)? The data shown in Figure 5 for example are corrected for truncation, and if yes, how? And what about the uncertainty?

Section 5: together with the asymmetry factor is also possible to retrieve the lidar ratio at the used wavelength of 532 nm? If yes, I would suggest to do it. The lidar ratio is a useful parameter to provide as output.

---

## Referee Comment (RC2) · Anonymous Referee #2 · 17 Jul 2018

The manuscript presents field measurements of the volume scattering function of various atmospheric aerosols including monitoring of several Saharan dust outbreaks over Europe. The measurements are performed at 532 nm spanning a scattering angle range from 5 to 175 degrees. The polar nephelometer is located at 2500 m high in the Sierra Nevada, Spain. The corresponding scattering coefficients are apparently simultaneously measured with a three wavelength integrating nephelometer. The results presented in the paper are of interest for the atmospheric community. However some missing information must be provided before publication. Short and concise papers are much appreciated but they should provide sufficient information so that they are fully understandable.

General comments

- The introduction is extremely short. It should at least include information (scope,

instruments involved, locations, etc.) on the SLOPE campaign to which the measurements belong. Also some annual statistic of Sahara sand dust storm over southern Europe would be appreciated.

- Information and proper references are provided for the custom made polar nephelometer but no information at all is given for the (commercial?) integrating nephelometer. Please, include in section 3 (instrumentS and method) basic information for the integrating nephelometer. That would help in understanding e.g. the text in page 6, lines 5-10. How are BbsG and BsG defined ?

- As mentioned in the text the measurements are performed in a certain time period but not continuously. Please, provide a table with detailed information on the July 2016 Sierra Nevada campaign: instruments (nephelometer/integrating nephelometer), dates, sample time. It would also be interesting to combine the time table of the measurements with the information on the back trajectories from NOAA (current Table 1).

Section 4.

- Results: How are period 1, 2 and, 3 defined?

- According to Figure 4 it seems like simultaneous measurements with the integrating and polar nephelometer are obtained. However, in Figure 5 the measured integrated volume scattering functions are obtained in narrower time periods. Please, clarify.

Minor comments:

- Page 3, first paragraph, 2nd line: "alpha <-1" should be "alpha < 0".

- Page 3, last paragraph, second line: "Using data given by given by" should be "given by"

- Page 5, line 29: "In figure 5.."do you mean in Figure 4?

---

## Author Comment (AC1) · 13 Aug 2018

My comments to referee 2

Q: General comments - The introduction is extremely short. It should at least include information (scope instruments involved, locations, etc.) on the SLOPE campaign to which the measurements belong. A: The instruments are described in the revised version as well as the SLOPE campaign. The location (Albergue Universitario of the University of Granada, located in the Sierra Nevada at an elevation of 2505 m a.s.l. Its coordinates are 37° 5' 43.72"N, 3° 23' 12.57"W) is already described on page 4 of the paper under discussion.

Q: Also some annual statistic of Sahara sand dust storm over southern Europe would be appreciated. A: I have added the total days and the total number of Sahara dust events, from May to September 2016, as well as a reference in a footnote. Q: - In-

formation and proper references are provided for the custom made polar nephelometer but no information at all is given for the (commercial?) integrating nephelometer. Please, include in section 3 (instrumentS and method) basic information for the integrating nephelometer. A: I have added three lines on the Integrating Nephelometer and a reference to the NOAA site. (Since NOAA uses this instrument since more than 20 years at their baseline stations, the instrument is thoroughly tested and competent instructions on calibration and evaluation can be found on the site)

Q:That would help in understanding e.g. the text in page 6, lines 5-10. How are BbsG and BsG defined ? A: A definition is given towards the end of section 5.

Q: - As mentioned in the text the measurements are performed in a certain time period but not continuously. A: First an explanation for the discontinuity of the measurements: Computer failure (I/O error) stopped the measurement and it had to be started manually. Since I stayed at Granada and was brought to the measuring site, no data could be obtained until I returned to the site. From June 17 to 24, 2016 I had to be at the University of Vienna and no measurements could be performed. The Integrating Nephleometer less frequently failed to work.

Q: Please, provide a table with detailed information on the July 2016 Sierra Nevada campaign: instruments (nephelometer/integrating nephelometer), dates, sample time. A: This actually can be seen in figure 5 of the paper under discussion. The red line gives the data of the Integrating Nephelometer, one can see a few periods of failure. The black dots are the measurements of the polar nephelometer, So I do not think it is not necessary to add an extra table, which does not give more information.

Q: It would also be interesting to combine the time table of the measurements with the information on the back trajectories from NOAA (current Table 1). A: This actually was already done in figure 4: at the top of the figure the classifications can be found.

Q: Section 4. - Results: How are period 1, 2 and, 3 defined? A: The choice of the periods was accidentally, due to failure and absence (see above). Unexpectedly in

these periods different aerosols dominated.

Q:- According to Figure 4 it seems like simultaneous measurements with the integrating and polar nephelometer are obtained. However, in Figure 5 the measured integrated volume scattering functions are obtained in narrower time periods. Please, clarify. A: Please note that in figure 4 the scale has two breaks in the x-axis, so periods, where no measurements with the polar nephelomenter could be performed are not shown. I have used this representation in order to show more details.

Q:Minor comments: - Page 3, first paragraph, 2nd line: "alpha <-1" should be "alpha < 0". A: Has been corrected Q:- Page 3, last paragraph, second line: "Using data given by given by" should be "given by" A: Has been corrected Q:- Page 5, line 29: "In figure 5.."do you mean in Figure 4? A:You are right this has been a mistake and is corrected.

I also upload the revised veraion of the paper as supplement

Please also note the supplement to this comment:
https://www.atmos-chem-phys-discuss.net/acp-2018-464/acp-2018-464-AC1-supplement.pdf

**Supplement:**

**Angular Scattering of the Sahara Dust Aerosol**

Helmuth Horvath[1], Lucas Alados Arboledas[2,3], Francisco Jose Olmo Reyes[2,3]

[1]University of Vienna, Faculty of Physics, Aerosol Physics and Environmental Physics, A-1090 Vienna, Austria
[2]University of Granada, Department of Applied Physics, E-18071 Granada, Spain
5  3Andalusian Institute for Earth System Research (IISTA-CEAMA), Granada, Spain

*Correspondence to*: Helmuth Horvath (Helmuth.Horvath@univie.ac.at)

**Abstract.** Soil erosion aerosols can be transported considerable distances, the Sahara is one of the major sources on the world. In June 2016 the volume scattering function of the atmospheric aerosol has been determined in the Sierra Nevada, Spain, at an altitude of 2500 m. Measurements were performed with a polar nephelometer permitting measurements
10 between scattering angles of 5° to 175°. The values at the missing angles could be estimated to a high accuracy, using the shape of the scattering function adjacent to the missing angles, thus a complete volume scattering function was available. During the measuring period intrusions of long range transported Sahara aerosol happened several times. The classification of the aerosol was done by back trajectories and by the Angström exponent of the wavelength dependent scattering coefficient, which was determined by a three wavelength Integrating Nephelometer. The phase function of the Sahara
15 aerosol had a stronger forward scattering,  and less backscattering compared to the non-Sahara aerosol, which is in agreement with other findings for irregular particles. The asymmetry parameter of the phase function is the best characteristic to distinguish Sahara Aerosol from non-Sahara aerosol. In this study the asymmetry parameter for the Sahara aerosol was larger than 0.65, whereas the non-Sahara aerosol had an asymmetry parameter below 0.6. A comparison with measurements performed with long range transported Gobi Desert aerosols observed in Kyoto, Japan, showed very
20 similar results.

**1. Introduction**

Deserts are a major source of aerosol particles. On a global scale the desert aerosol contributes 60–1800Tg/y of the total
25 yearly aerosol production of 2900–4000 Tg (Jaenicke, 1988). Junge (1979) estimates the global source strength of deserts as 260 to 400 Tg/y, with the Sahara contributing 60 to 200 Tg/y. IPCC (2001) estimates the yearly mineral dust emissions as 1000 to 3000 Tg/y, amounting to 1/3 to ½ of the global emissions. In contrary to most of the other particles present in the atmosphere, desert aerosol particles are produced from minerals by mechanical processes, which obviously lead to irregular shaped particles. This is documented by innumerous electron micrograph studies (see e.g. Falkovich et al. 2001, Iwasaka et
30 al., 2003, Kandler et al, 2006). Among many other effects, the desert aerosol is expected to have an influence on the radiative balance, see e.g. Obregón (2014), Valencuela et al. (2012), Antòn et al. (2014).

The mechanical production of particles results in particle sizes usually larger than 1 μm. Mean sizes range up to 10μm (Kontratyev et al. (2006), Alfaro et al. (2003), Cheng et al. (2005), Xin et al. (2005), Alfaro & Gomez, (2001), Falkovich et al., (2001)), and during atmospheric transport the size distribution is modified, mainly due to the shorter life time settling of of larger particles: The average residence time in the troposphere is estimated as 10 days for 1 μm particles and 3 days for 10 μm particles. (Jaenicke, 1988)

Since the dust particles have an irregular shape, their scattering properties are difficult to model, but their scattering properties are urgently needed to estimate the effect on climate. Below we report measurements of the volume scattering function of the Sahara aerosol transported to the Iberian Peninsula, which were performed during several dust outbreaks.

**2 Definitions, units, and nomenclature**

The volume scattering function $\gamma(\theta)$ of the aerosol is defined as follows, see figure 1: Let a volume dV of aerosol be illuminated by radiation having a flux density S. The light flux dΦ scattered into a solid angle dω at the scattering angle θ is obtained as

[revised manuscript text omitted]

---

## Author Comment (AC2) · 13 Aug 2018

Q:In general, the paper misses from a formal uncertainty analysis of measured and derived parameters. Also, uncertainties are missing in all plots. Please correct this and add more formal error discussion. For example, in Figure 7 of the paper we can see differences between the dust and non-dust cases, but what is the real difference within uncertainties? A:In figure 7 the standard deviation of the average of the Sahara and Non_Sahara data has been added. It is obvious that a significant difference between the two datasets exists. Doing this I realized that I plotted the wrong average for the Sahara aerosol. Instead of copying the average I copied the line above, which was the last measurement of the Sahara aerosol

Q:Specific comments Page 1, line 29: please check the extra comma in the text: A: has been corrected

[Figure]

Q: Page 2, line 4: add references after "larger particles" A: I have added a short discussion of the average residence time of the particles, including a reference

Q: Page 2, line 12: E should be replaced by S, I guess A: This was a mistake, thank you for pointing it out.

Q: Page 2, line 30: please check the formula since I think you missed a minus sign before the angstrom exponent; if I am right, then check the following discussion. A: There are two ways to use the Angström formula: $\sigma(\lambda) = \sigma(\lambda 0)Å\mathring{u}( \lambda/\lambda 0)\alpha$ or $\sigma(\lambda) = \sigma(\lambda 0)Å\mathring{u}( \lambda/\lambda 0)\text{-}\alpha$ I have used the first possibility, which historically was used earlier. It is used consistently in the paper.

Q: Page 3, line 5: I do not like the expression "usual aerosol", please be more specific (pollution aerosols, fine aerosols?). A: I have replaced "usual aerosol" by "non-Sahara aerosol"

Q: Page 3 line 5; I would replace with "is a sign for desert aerosol particles" with "it is a sign for large aerosols, as desert dust" or similar A: I have changed it to "an aerosol containing larger particles, in this study mainly desert particles"

Q: Page 3, line 25: how the extrapolation is done? A: The method is described in detail in the reference (Horvath, 2015) given in the paper under discussion. Since this reference is a 10 page paper I can only give a short description: Analyzing a large set of scattering functions of both spherical and non-spherical particles it was found, that it is possible to predict the shape of the scattering function for a few degrees ahead if the shape of the curve is known up to the point, where the extrapolation starts. Since only 5 degrees are missing this can be done quite accurately.

Q: Page 4, line 5: you refer to "all the instruments", which instruments? Please describe clearly the instruments used. I am also a bit confused by the fluxes. A flux of aspiration for the custom made nephelometer is specified in the previous page, while here there is reference to a different flow rate. What is this for? Q: The SLOPE study mainly

was intended to determine the vertical structure of the aerosol by remote sensing instruments. So the main instruments were sun and sky photometers and an airplane. Obviously ground based instruments were also used. At the Albergue Universitaria, several instruments were operated. It is a standard practice to connect these instruments to a central sampling port. This is a vertical tube extending above the roof, through which air is sucked into the laboratory by a blower. The flow rate is chosen such that as little as possible disturbances of the aerosol take place, thus it can be assumed that the instruments sample undisturbed outside air. I have listed the other instruments, although they are irrelevant for this study.

Q: Page 4, line 12: again there is the expression "usual aerosol" to modify A: Has been replaced by non-Sahara aerosol

Q: Page 4, line 19-22: the integrated nephelometer mentioned in this paragraph was not introduced before. Please, again, clearly indicate the used instruments and their configuration. A: I have listed the other instruments although they are irrelevant for this study.

Q: Moreover, what about the integrating nephelometer (model, data treatment, uncertainties)? A: I have added three lines on the Integrating Nephelometer and a reference to the NOAA site. (Since NOAA uses this instrument since more than 20 years at their baseline station, the instrument is thoroughly tested and competent instructions on calibration and evaluation can be found on the site)

Q: The data shown in Figure 5 for example are corrected for truncation, and if yes, how? A: In figure 5 no truncation procedure is needed to apply. The calibration of the Integrating Nephelometer is done according to the NOAA instructions and compared to the integrated volume scattering function of the polar nephelometer, which should be identical.

The effect of truncation is shown in Figure 9 and I have added an explanation how the signals BsbG and BsG are obtained, when using the measured volume scattering

function.

Q: And what about the uncertainty? A: This question is difficult to answer. Under laboratory conditions i.e. when a constant aerosol is produced e.g. by a constant output atomizer, the Integrating Nephelometer measures a signal which is constant as long as the atomizer is in operation. Similarily the polar nephelometer measures identical volume scattering functions, which, when plotted on top of each other are one line. So the uncertainty of both instruments is 2% or even better. BUT the atmosphere is not laboratory with a constant aerosol, especially at the site in the Sierra Nevada with a layered aerosol. This can best be seen in Figure 5. The continuous line is the scattering coefficient of the aerosol (if the aerosol were constant the instrument would produce a horizontal line). So the aerosol is variable and the instruments measure this variable aerosol with very little uncertainty. The variability of the aerosol can best be seen in figure 7 and Table 2. For the aerosols classified as Sahara, the phase function at 90° e.g. is 0.35 sr-1 with a standard deviation of 0.04sr-1 or 11%. But this variability is not caused by an imprecise instrument, but by an aerosol which just is not constant.

Q: Section 5: together with the asymmetry factor is also possible to retrieve the lidar ratio at the used wavelength of 532 nm? If yes, I would suggest to do it. The lidar ratio is a useful parameter to provide as output. A: The lidar ratio is defined as the extinction coefficient divided by volume scattering coefficient at 180°, or $4\pi$ /([P(180°) .$\omega$], with $\omega$ the single scattering albedo. I have used an average value for $\omega$ and the value for the lidar ratio is listed in table 2. I have appended teh revised version of the paper

Please also note the supplement to this comment:
https://www.atmos-chem-phys-discuss.net/acp-2018-464/acp-2018-464-AC2-supplement.pdf

---

## Author Response (AR2)

Dear Editor!

November 26, 2018

Thank you for the suggestions, as you can see I have altered the introduction considerably and added one point to the conclusion.

I have loaded the manuscript with markups as manuscript and the manuscript without markups as supplement.

November 27, 2018

As pointed out by Svenja Lange I have added two sentences on authors' contributions

Sincerely,

H. Horvath

[revised manuscript text omitted]

Figure 2

**Figure 2: Principle of the polar nephelometer**

[Figure]

Figure 3

**Figure 3: Back trajectories used for classification of aerosol types**

[Figure]

**Figure 4: Angström exponent of the spectral scattering coefficient of the aerosol measured and classification by back trajectories. For the desert aerosol the Angström Exponent is larger than -1. In addition the asymmetry parameter is plotted too, for desert aerosol it is larger than 0.65.**

[Figure]

**Figure 5: Measured scattering coefficient of the aerosol during the three periods of observation. The solid red line is the signal of the Integrating Nephelometer, the points are scattering coefficients obtained by integrating the measured volume scattering function.**

[Figure]

**Figure 6. Comparison of phase function attributed to Sahara and non-Sahara aerosols.**

[Figure]

**Figure 7. Average of the Sahara and non-Sahara aerosol phase functions.**

[Figure]

**Figure 8. Relationship between the measured Angström exponent and the asymmetry parameter using the measured phase function.**

[Figure]

Back_period3

**Figure 9. Back scattered fraction measured with the Integrating Nephelometer (red curve) and values obtained by the measured Volume scatteing function. Simulation of the truncation is shown by the hollow points.**

[Figure]

AsybackAerosolTyps_with_Gobi

**Figure 10: Asymmetry parameter and backscattered fraction calculated for the aerosols measured during this study. A clear separation between Sahara and non-Sahara aerosols is evident. The data points are in excellent agreement with data for the Gobi Desert aerosol.**

| Table 1. Classification of back trajectories | | |
|---|---|---|
| Name | Air passes over | Remark |
| „Atlantic" | 60°W<longitude<0°, 35°N<latitude<40°N | |
| "Atlantic North" | 40°W<longitude<0°, 60°N<latitude<40°N | |
| "Sahara" | 10°W<longitude<5°E,37°N<latitude<26°N | |
| "Sahara high" | 5°W<longitude<5°E, 37°N<latitude<26°N | |
| "Sahara/Atlantic", "Atlantic/Sahara" resp. | Up to 500 km West of the coast of Africa | Distinction between the two by Angström exponent |
| "North Africa/Mediterranean" | 6°W<longitude<8°E, 35°N<latitude<37° | |

| Table 2. Characteristics of the two types of aerosols | | | | |
|---|---|---|---|---|
| | Sahara aerosol | | Non-Sahara Aerosol | |
| | Value | Stddev. | Value | Stddev. |
| Asymmetry parameter  [-] | 0.71 | 0.03 | 0.56 | 0.05 |
| Backscattered fraction  [-] | 0.094 | 0.014 | 0.153 | 0.027 |
| Average scattering coefficient [Mm$^{-1}$] | 71 | 22 | 22.3 | 7.3 |
| Phase function at 0°  [sr$^{-1}$] | 67 | 27 | 31 | 10 |
| Phase function at 90°  [sr$^{-1}$] | 0.26 | 0.03 | 0.35 | 0.04 |
| Phase function at 180°  [sr$^{-1}$] | 0.31 | 0.11 | 0.59 | 0.21 |
| Lidar ratio [sr$^{-1}$] | 44 | 16 | 23 | 8 |

---

## Author Response (AR3)

Authors response:

November 29, 2018

I have removed the typos mentioned (and found some more, which are removed also) And defined PM10 and PM20 in the text.

H. Horvath